# Liver Graft Hypothermic Static and Oxygenated Perfusion (HOPE) Strategies: A Mitochondrial Crossroads

**DOI:** 10.3390/ijms23105742

**Published:** 2022-05-20

**Authors:** Raquel G. Bardallo, Rui T. da Silva, Teresa Carbonell, Carlos Palmeira, Emma Folch-Puy, Joan Roselló-Catafau, René Adam, Arnau Panisello-Rosello

**Affiliations:** 1Department of Cell Biology, Physiology and Immunology, Universitat de Barcelona, 08028 Barcelona, Catalonia, Spain; rgomezbardallo@ub.edu (R.G.B.); tcarbonell@ub.edu (T.C.); 2Center for Neuroscience and Cell Biology, Universidade Coimbra, 3000-370 Coimbra, Portugal; rtsilva@cnc.uc.pt (R.T.d.S.); palmeira@ci.uc.pt (C.P.); 3Experimental Pathology Department, Institut d’Investigacions Biomèdiques de Barcelona (IIBB), Spanish National Research Council (CSIC)-IDIBAPS, CIBEREHD, 08036 Barcelona, Catalonia, Spain; emma.folch@iibb.csic.es (E.F.-P.); jrcbam@iibb.csic.es (J.R.-C.); 4Centre Hépato-Biliaire, AP-PH, Hôpital Paul Brousse, 94800 Villejuif, France; rene.adam@aphp.fr

**Keywords:** liver graft preservation, AMPK, succinate, ALDH2, glycocalyx

## Abstract

Marginal liver grafts, such as steatotic livers and those from cardiac death donors, are highly vulnerable to ischemia–reperfusion injury that occurs in the complex route of the graft from “harvest to revascularization”. Recently, several preservation methods have been developed to preserve liver grafts based on hypothermic static preservation and hypothermic oxygenated perfusion (HOPE) strategies, either combined or alone. However, their effects on mitochondrial functions and their relevance have not yet been fully investigated, especially if different preservation solutions/effluents are used. Ischemic liver graft damage is caused by oxygen deprivation conditions during cold storage that provoke alterations in mitochondrial integrity and function and energy metabolism breakdown. This review deals with the relevance of mitochondrial machinery in cold static preservation and how the mitochondrial respiration function through the accumulation of succinate at the end of cold ischemia is modulated by different preservation solutions such as IGL-2, HTK, and UW (gold-standard reference). IGL-2 increases mitochondrial integrity and function (ALDH2) when compared to UW and HTK. This mitochondrial protection by IGL-2 also extends to protective HOPE strategies when used as an effluent instead of Belzer MP. The transient oxygenation in HOPE sustains the mitochondrial machinery at basal levels and prevents, in part, the accumulation of energy metabolites such as succinate in contrast to those that occur in cold static preservation conditions. Additionally, several additives for combating oxygen deprivation and graft energy metabolism breakdown during hypothermic static preservation such as oxygen carriers, ozone, AMPK inducers, and mitochondrial UCP2 inhibitors, and whether they are or not to be combined with HOPE, are presented and discussed. Finally, we affirm that IGL-2 solution is suitable for protecting graft mitochondrial machinery and simplifying the complex logistics in clinical transplantation where traditional (static preservation) and innovative (HOPE) strategies may be combined. New mitochondrial markers are presented and discussed. The final goal is to take advantage of marginal livers to increase the pool of suitable organs and thereby shorten patient waiting lists at transplantation clinics.

## 1. Introduction

In liver transplantation, graft quality is the key factor that determines procedural success and long-term survival and outcome, but this depends not only on its intrinsic performance but also on the time and quality of preservation and transportation from donor to recipient implantation. At Christmas in 1971, Belzer decided to ship a recovered kidney in San Francisco (USA) to the Leiden transport for transplantation. After 37 h of in-flight preservation, the kidney was successfully transplanted into a 42-year-old truck driver with polycystic kidney disease, as planned [1,2].

Since the first investigations initiated by Belzer et al. [1,3,4,5,6,7], much progress has been made in the preservation of liver grafts [8,9,10]. The actual challenge in liver transplantation is organ scarcity and the pressing need for liver transplantation, which have led physicians to use marginal livers, such as steatotic livers, to increase the organ pool for transplantation [11,12,13,14]; however, its high vulnerability against ischemia and reperfusion injury [14,15,16] due to microcirculation exacerbated by fat accumulation in hepatic sinusoids could compromise graft viability and further outcomes with post-transplantation problems [17,18,19].

Cold static preservation is a mandatory step characterized by oxygen deprivation to the graft that provokes ischemic damage, leading to mitochondrial integrity changes and energy metabolism breakdown alterations that affect fatty liver graft integrity during cold storage in experimental settings. Fatty livers exhibit reduced tolerance against ischemic events, with further reduced ATP levels and greater injury levels when compared to nonsteatotic livers [20,21]. However, the cold ischemic time accepted for clinical purposes is 12 h [22,23], considered suitable against the apparition of “primary nonfunction”, graft failure, and patient death, along with reduced long-term graft survival [22].

The mechanism of mitochondrial metabolic changes upon cold ischemia is relatively well known [24], as shown in Figure 1. During prolonged cold graft storage, the oxygen deprivation conditions allow diminishing the metabolic demands, and consequently, the mitochondrial respiration complexes are seriously altered, leading to succinate accumulation at the end of ischemia [24]. Succinate dehydrogenase (SDH) activity drives a significant portion of ROS generation, and it has been demonstrated that SDH inhibition by malonate is protective against reperfusion injuries. Notably, itaconate inhibits SDH in a dose-dependent manner, leading to succinate accumulation [25,26], and its exogenous administration modifies the host response to ischemia–reperfusion that is sufficient to suppress reperfusion-related injuries [24,25,26,27,28].

The accumulation of succinate is partly responsible for controlling graft injury after oxygenation [24,25,26,27,28], and consequently, the ability of itaconate to inhibit succinate dehydrogenase in complex II may also play a determining role, reducing the initial succinate metabolism after earlier oxygenation before reperfusion conditions (Figure 1) [24,25,26,27,28]. Moreover, the longer the ischemic period lasts, such as during liver graft cold storage, the more compromised the mitochondrial antioxidant electron transport chain (ETC) system, depleting substrates such as glutathione, which render the cells more susceptible to oxidative stress at reperfusion after graft revascularization.

With this in mind, it seems necessary to envisage new directions in graft cold static preservation strategies [8,29], possibly in combination with newer techniques developed, such as hypothermic oxygenated perfusion (HOPE) using machine devices [30,31], for better protection from harvest to revascularization. New protective strategies to increase graft protection are urgently needed to take advantage of marginal grafts and increase the donor pool to reduce waiting lists for liver transplantation. This review covers different aspects of hypothermic liver graft preservation as follows: cold ischemic insult and liver graft cold storage are discussed in Section 2; mitochondrial protection and organ preservation solutions in static cold storage are discussed in Section 3; new additives for improving cold static preservation are discussed in Section 4; HOPE, mitochondrial protection, glycocalyx preservation, and PEG35 effluents (IGL-2) are discussed in Section 5; and (5) some considerations and concluding remarks are presented in Section 6.

## 2. Cold Ischemic Insult and Liver Graft Cold Storage

Crucial adverse consequences for liver graft during cold storage due to deprived oxygen conditions have led us to use organ preservation solutions such as UW, HTK, Celsior, and IGL-1 [8,29]. UW and IGL-1 solutions are characterized by an oncotic agent in their composition such as hydroxyethyl starch (HES) in UW and PEG35 in IGL-1 [8,29], in contrast to HTK and Celsior, which do not contain an oncotic agent in their formulations [8]. Moreover, IGL-2 showed higher concentrations of PEG35 and glutathione compared to IGL-1. In clinical transplantation, IGL-1, Celsior, and HTK were demonstrated to be suitable alternatives to UW (gold-standard reference), although some limitations for HTK were recently described in the European Liver Transplantation Register (ELTR) [32].

Since 2006, we have investigated the protective mechanisms exerted by polyethylene glycol 35 (PEG35) in IGL-1 solution, which was found suitable for fatty liver graft preservation [33]. In this line, the presence of PEG35 in IGL-1 [33] and rinse solution [34] was a determinant for protecting the graft against the deleterious effects accumulated from organ recovery and washout up to graft cold storage and the following reperfusion [33,34]. This confirms that PEG35 is a useful tool for preventing cold IRI associated with transplantation [35]. The benefits of PEG35 in rinse and IGL-1 solutions are mainly associated with the prevention of mitochondrial damage, activated protective cell signaling mechanisms such as AMPK (energy sensor), and nitric oxide generation through constitutive eNOS activation [33,34,36,37].

With this in mind, we recently proposed IGL-2 [38] in order to enhance fatty liver graft protection by preserving liver damage with mitochondrial integrity inherent in oxygen deprivation conditions during static cold graft storage. As shown in Figure 2, IGL-2 protected liver injury more efficiently (measured as AST/ALT transaminases) and mitochondrial damage (measured as GLDH) than UW and HTK solutions.

## 3. Mitochondrial Protection and Preservation Solutions in Static Cold Storage

ATP energy metabolism breakdown is inherent to oxygen deprivation conditions during graft cold storage. As shown in Figure 3, noticeable prevention of ATP breakdown after 24 h cold storage occurred when IGL-2 was compared to UW and HTK.

Moreover, ATP energy metabolism breakdown is associated with alterations in mitochondrial respiration complexes that occur as a consequence of oxygen deprivation, forcing the cell to adapt to anaerobic metabolism that leads to the accumulation of metabolites such as succinate (Figure 1), which will partly control graft damage when oxygen restoration occurs after reperfusion [24]. As shown in Figure 3, IGL-2 solution was capable of limiting succinate accumulation in a more efficient way compared to UW and HTK. The presence of PEG35 and glutathione in IGL-2 contributed to modulating succinate accumulation at the end of cold ischemia. Other metabolites such as itaconate, which accumulated at the same time, may partially contribute to inhibiting succinate dehydrogenase (SDH) during earlier stages of reperfusion [26,27,28]. Consequently, both succinate and itaconate levels accumulated at the end of the cold storage period will have an important role in determining the viability of the graft after revascularization.

It is well known that fatty livers show reduced tolerance to ischemic events with further reduced ATP levels and a greater injury level compared to nonsteatotic livers [21]. It has been discussed that mitochondrial uncoupling protein-2 (UCP-2), highly expressed in steatotic hepatocytes, may be responsible for the aforementioned higher fatty liver sensitivity to ischemia [39]. UCP-2 acts as an inner-mitochondrial membrane proton carrier that uncouples ATP synthesis, facilitating proton leakage into the mitochondrial matrix and uncoupling mitochondrial respiration. Increased UCP-2 expression was associated with low ATP levels in hearts preserved in Celsior solution after static hypothermic preservation [40]. This would suggest that UCP-2 has a relevant role between mitochondrial machinery function and ATP preservation during cold graft storage [40]. The use of genipin, a UCP-2 inhibitor added to Celsior solution, would contribute to limiting ATP depletion and protecting fatty livers against cold ischemic insult during IRI [40]. UCP inhibitors, such as 2,4-dinitrophenol and others, have been proposed, but no clinical applications have been carried out [41,42].

Mitochondrial aldehyde dehydrogenase-2 (ALDH2) is best known for its critical detoxifying role in liver alcohol metabolism, but there is growing evidence that it also plays a role in IRI through the prevention of oxidative stress processes (oxygen free radicals and toxic 4-hydroxy-nonenal generation) [20,43]. PEG35 in rinse [34] and IGL-1/IGL-2 solutions [33,44] was a determinant for increasing mitochondrial protection and function, as well as preventing lipid peroxidation when compared to identical solutions with or without PEG35 [44]. The high antioxidant capacity of IGL-2 was corroborated by MDA and AOPP when compared to livers preserved in UW and HTK (Figure 4).

Although oxygen levels of the media embedding the organ (preservation solution) are very low, there is still an impairment of mitochondrial oxidative phosphorylation, leading to the generation of reactive oxygen species (ROS). In this sense, NO generation in IGL-1/IGL-2 solutions [33,38] could act, among many other functions, as an ROS scavenger in the lipoperoxidation process [44]. Interestingly, these involved cytoprotective mechanisms observed for PEG35 in IGL-1 and IGL-2 solutions were similar to those we proposed for liver ischemic preconditioning strategies in a rat model, where AMPK and NO are also well-known liver preconditioning cytoprotective factors [36,37].

With this in mind, we consider that preservation solutions can be considered preconditioning tools against IRI. This NO generation and its vasodilator properties are especially beneficial for steatotic liver grafts, in which fat accumulation in the sinusoidal space exacerbates microcirculation alterations [16,45].

## 4. New Additives for Improving Cold Static Preservation: Oxygen Carriers, Ozone, and AMPK Inducers

### 4.1. Oxygen Carriers: M-101

Static hypothermic preservation of liver grafts static hypothermic preservation implies a lack of oxygen, and its adverse consequences affect energy metabolism, blocking mitochondrial respiration, concluding with succinate accumulation. Under these circumstances, the use of a natural extracellular oxygen carrier added to the preservation solution could help maintain the functionality of certain metabolic pathways, hence avoiding the nocive accumulation of succinate.

M-101 is a natural giant extracellular hemoglobin (Hb) extracted from the marine invertebrate *Arenicola marina* that was first used by Alix et al. as an additive to UW solution in pig liver transplantation [46]. The authors demonstrated that UW solution enriched with M-101 improved liver graft oxygenation when compared to simple UW solutions, but it did not reach the oxygenation level achieved with alternative dynamic hypothermic oxygenated machine perfusion (HOPE) strategies (with active flux of oxygen) [46]. This transient restoration of oxygenation achieved with M101 presumably contributes to maintaining liver mitochondrial machinery to function at basal levels, thus avoiding the accumulation of succinate during cold ischemia, according to Schlegel’s mechanism [25]. Furthermore, the M-101 additive is also useful for fatty liver static preservation when combined with HOPE [47], but additional studies should investigate in depth its usefulness in maintaining graft quality when longer cold storage periods are applied, especially in vulnerable organs (DCD donors, pancreas, and fatty liver grafts).

### 4.2. Ozone

Ozone is a gas with antioxidant effects [48,49] that has been proposed as an additive in UW solution (gold-standard reference) to minimize cold ischemic damage during graft preservation in UW solution [50]. Ozone protection during cold storage modulates the XHD/XOD system, preserving adenosine storage and blocking the xanthine/xanthine oxidase pathway that promotes ROS generation [51]. Taking this into account, ozone persufflation strategies should be considered and further evaluated as a potential tool for hypothermic graft preservation strategies.

The deleterious oxidation consequences of oxygen generated by ozone conversion to oxygen during cold storage would be presumably even more negligible when glutathione and allopurinol are present in the composition of UW and IGL-2 solutions. Presumably, the persufflation of the liver graft before cold graft storage in ozonized UW solution could offer additional protection to preserve the liver graft.

### 4.3. Other Antioxidants

It is well known that glutathione is present in its reduced form and is a critical component often used in commercial UW and IGL-1 [8]. Reduced glutathione is a very labile component, and its auto-oxidation is accelerated by an increase in temperature and contact with oxygen [52]. With this in mind, we assume that long periods of storage at room temperature are not recommended for storing any preservation solutions containing glutathione that could affect the initial glutathione composition. [53]. This was one reason why we proposed increasing the content of glutathione in IGL-2 solution [44] vs. UW and IGL-1 [33]

In addition, well-known hydrosoluble antioxidants (including vitamin C, N-acetyl cysteine (NAC), alpha-tocopherol analog, and others) were used as additives to preservation solutions to reinforce their respective antioxidant features [54,55,56,57]. Further investigations are needed to explore new antioxidants for graft preservation purposes.

### 4.4. Adenosine Monophosphate Protein Kinase (AMPK) Inducers

Graft oxygenation is an alternative to solve the hypoxic conditions during cold storage; however, other studies focus on another of the main problems occurring during ischemia, such as the breakdown of energy metabolism and the dropping of ATP levels, which can be countered by the activation of the AMPK energy sensor [58,59,60]. In this line, AMPK inductors, such as 5-aminoimidazole-4-carboxamide ribonucleotide (AICAR) or metformin, have evidenced their benefits when added to UW solutions [61,62]. Other natural AMPK activators, such as 3,5,4′-trihydroxy-*trans*-stilbene (resveratrol) [63], could be good candidates as additives for preservation solutions in future approaches. This principle also applies to PEG35 solutions, in which there has been an increase in activated AMPK under hypoxic conditions [64,65,66]. The mechanisms by which PEG35 activates AMPK in IGL-1 and rinse solutions remain to be investigated in depth.

## 5. HOPE, Mitochondrial and Glycocalyx Protection, and PEG35 Effluents (IGL-2)

The introduction of dynamic strategies by using machine devices for graft conservation, such as hypothermic oxygenated perfusion (HOPE), is a promising tool to not only increase graft quality for transplantation [67,68,69,70,71,72] but also to make useful marginal organs that were originally discarded for transplantation [31].

The use of transient and sustained oxygenation to the graft is the “key” that maintains the operational mitochondrial machinery at basal levels during HOPE, whose protective mechanisms were described by Schlegel and Cols [25]. Transient graft oxygenation during HOPE activates the mitochondrial machinery through mitochondrial complexes I and II, preventing preservation. Therefore, the importance of succinate metabolism by complex II/SDH during HOPE was shown to slowly re-establish a normal directed electron flow without significant ROS release, thereby slowing the breakdown of accumulated succinate (Figure 1) [24,25,26,27].

The mechanisms proposed by Schlegel et al. [25] also apply to static preservation, but in this case, the benefits of decreasing the accumulation of succinate can only be altered by preservation solutions, which can do so to a very limited extent. This is the case of IGL-2 solution, which better prevented succinate accumulation when compared to HTK and UW, the gold-standard reference (Figure 3), bringing the graft to more suitable conditions for the subsequent HOPE application. In clinical and experimental settings, the most used perfusate used in HOPE is modified UW solution, named Belzer MP, adapted for dynamic perfusion and maintaining HES as an oncotic agent in its formulation [25,67]. It is well known that HES can induce hyperaggregability of red blood cells [73], this being a limitation for graft washout and cold storage. This serious issue is not present with other oncotic agents such as PEG35 [74], which also confers more mitochondrial protection than HES.

Fluid dynamics constitutes one of the most differential features when HOPE and cold static preservation are compared. In any kind of dynamic perfusion, the perfusate applies pressure and shear stress that may involve the destruction of the superficial sugar thin layer covering the endothelium, known as “glycocalyx” [75,76]. The presence of PEG35 in perfusates (IGL-2 solution) contributes to favoring mechanotransduction processes inherent to fluid dynamics in HOPE, thereby also favoring better protection of the luminal glycocalyx [77]. Perfusates in a dynamic system, such as HOPE, can promote glycocalyx destruction, but this will greatly depend on factors such as viscosity, pressure, and other properties of the perfusate, which in turn depend on the components of the preservation solution, such as the oncotic agent, whose characteristics can be delimited by factors such as temperature [77].

Notably, PEG35 in IGL-2 lowers the viscosity in Belzer MPS currently used in HOPE by half [67]. The properties of PEG35 in effluents for HOPE and its effects in the organ justify a deep debate to change the paradigm on which is the better approach for dynamic preservation considering the aforementioned factors (high mitochondrial protection, diminished viscosity, no red blood cell hyperaggregability) that have been overlooked since the beginning of dynamic perfusion. Furthermore, PEG35 seems to be a more efficient agent for the protection of mitochondrial machinery and should be considered as a main strategy for hypothermic graft conservation in the complex route of the liver graft from “harvest to revascularization”.

Recently, aldehyde dehydrogenase-2 has been described as a regulator of mitochondrial machinery during hypothermic machine perfusion to mitigate the deleterious effects of renal ischemia–reperfusion [78]. ALDH2 plays an important cytoprotective role in HMP, reducing the accumulation of 4-HNE and regulating the Akt/mTOR autophagy pathway in a similar way as in cold static preservation [79]. In addition, our previous observations in HOPE studies are consistent with better mitochondrial protection and increased ALDH2 in livers preserved for 7 h in IGL-2 after 1 h in HOPE and then 1 h of reperfusion with previous HOPE [67].

Taking this into account, mitochondrial GLDH and ALDH2 could be considered good indicators in HOPE to evaluate graft liver mitochondrial integrity and mitochondrial machinery function. This could also be presumably extended to cold static preservation enriched with oxygen inductors such as M-101, which could be combined with HOPE, as suggested by Alix et al. [46].

## 6. Some Considerations and Concluding Remarks

The challenge for the next decade is to develop technical strategies to rescue marginal organs. To do so, part of the effort should be aimed at mitochondrial protection and, therefore, at maintaining the mitochondrial machinery at basal levels to prevent energy breakdown and the subsequent generation of toxic metabolites, such as 4HNE. Certainly, HOPE is more efficient than static preservation because it provides the cell with an environment closer to physiological conditions, hence triggering less stress and coping mechanisms that inevitably lead to cell death. The continuous supply of oxygen, nutrients, and metabolic substrates for ATP generation and assuring the maintenance of mitochondrial machinery, which includes regulatory enzymes such as ALDH2, will in turn be proactive in the upregulation of other cytoprotective factors.

In HOPE, as mentioned above, one of the most common tools used is Belzer MP solution, containing an oncotic agent (HES) that provokes red blood cell aggregation [73], which can be fatal, and has some nonoptimal physical properties (high viscosity) that have been considered as a major factor of impact on shear stress occurring in the dynamic fluids system.

In this line, the use of PEG35 would be “killing two birds with one stone”, avoiding red blood cell hyperaggregation and bringing the solution to more adequate parameters of viscosity in order not to damage the endothelial cells and their associated structures, such as glycocalyx, its application in the narrowed sinusoids of steatotic liver being especially interesting [16,45]. Furthermore, these benefits are not only applied at the physical level, as the use of IGL-2 confers higher mitochondrial protection and assures better conditions for the ALDH2 enzymatic function to prevent oxidative stress in hypothermic graft preservation [43,44].

The MP benefits of ALDH2 extend further, as demonstrated by Lin et al. [78], who evidenced that ALDH2 regulated autophagy via the Akt/mTOR pathway to mitigate renal ischemia–reperfusion injury and can activate the negative regulatory mammalian target of rapamycin (mTOR) pathway. We established similar protective autophagic mechanisms when IGL-2 and IGL-1 solutions were used [38,79]. Moreover, the fluid dynamics existing in HOPE may have adverse consequences due to the viscosity of the effluent generally used, Belzer MPS, leading to endothelial glycocalyx destruction and thus altering the induced transduction mechanisms involved in HOPE when compared to PEG35 solutions [77,80].

In this sense, the use of IGL-2/PEG35 solution would be a better alternative than Belzer MP/HES by two means: (1) protecting the glycocalyx due to reduced viscosity that produces less shear stress, which is especially relevant in narrowed sinusoids and peripheral microcirculation [77], and (2) as a consequence of a less damaged glycocalyx, mechanotransduction mechanisms are more preserved, promoting NO generation (with vasodilator properties), being, again, beneficial in narrowed sinusoids [45,77].

The scarcity of organs and the growing demand for liver transplantation have led to hospitals using marginal grafts for organ transplantation. For years, static preservation has been based on the principles of hypothermic and static storage; however, the limitation of oxygen supply is serious for graft survival, given that the conditions proposed for static preservation are far from physiological. The use of HOPE is very promising [67,68,69,70,71,72], as well as gas additives or hemoglobins such as M-101 that, combined with further HOPE, could be an alternative method to palliate the deficient oxygenation of the graft during static preservation conditions [46,47].

Further investigations should be carried out on the usefulness of oxygen carriers, such as M101, for transporting organs in countries with complex logistic operations that extend the static cold storage periods, since currently, the dynamic systems known to provide oxygen in a dynamic system (HMP/NMP) are not sufficiently developed/cheap enough to be applied at a real clinical level.

With all this in mind, we assume that new promising predictors may evaluate graft quality at the end of the cold storage/cold ischemia step such as glutamate dehydrogenase (mitochondrial damage), ALDH2 (mitochondrial function), succinate and itaconate accumulations, and flavin mononucleotide (FMN); in contrast to HOPE, succinate and itaconate metabolites have a predictive value at early oxygenation/reperfusion times, including FMN [24,25,26,27]. Moreover, we could logically assume that the use of a unique solution, such as IGL-2 [81], could be an efficient and interesting tool to protect graft mitochondrial machinery and avoid the cumulative damage induced by the mandatory complex processes before transplantation, including organ recovery and its earlier washout, subsequent graft cold storage, and additional washout to finally proceed with HOPE strategies [25,67,68,69,70,71,72]. Certainly, the suitable use of a unique solution such as IGL-2 would simplify the logistics of liver transplantation, including also the combination of split liver transplantation with HOPE [82,83].

Finally, it is clear that new markers are needed to assess graft quality. In a recent editorial, Schegel et al. [84] asked whether the current combination of biomarkers used in liver transplantation could be useful in predicting liver function or whether we are only measuring actual lesions. In this sense, the accumulation of energy metabolites such as succinate and itaconate metabolites at the end of ischemia, together with GLDH and mitochondrial ALDH2 and UCP2, including energy sensors such as activated AMPK, could be promising predictors for the evaluation of the graft at the end of cold storage prior to reoxygenation, a phase in which the role of those metabolites is unleashed and in which the harm of ROS is a major issue.

### Summary

Mitochondrial preservation should be a prioritary direction for better conserving the graft when static and HOPE strategies are used in combination or alone. Promising markers such as GLHD, mitochondrial ALDH2, and accumulating succinate, itaconate, and flavin mononucleotide should be considered for future investigations in clinical transplantation. Glycocalyx and nitric oxide are altered in static preservation [80], but their use as predictors could be even more appropriate to HOPE strategies for evaluating the adverse effects of effluent dynamics and transduction mechanisms for future HOPE strategies, given the relevance of measuring glycocalyx alterations in clinical transplantation [85,86,87,88]. The benefit of a unique IGL-2 solution of static and dynamic HOPE preservation strategies is a topic that needs to be addressed for a simplified clinical procedure [81,83]. This will speed up current static and HOPE strategies in clinical liver transplantation, including liver split transplantation for HOPE application [82,83] and marginal livers. We assume that investigating in depth the underlying mitochondrial protection mechanisms [25,26] should be a priority direction for future approaches to liver hypothermic graft preservation strategies using PEG solutions. At this point, let us go back to 1989 when Belzer and Southard reported that the “mechanisms by which polyethylene glycols prevents cell swelling and thus maintains cell viability is not related to osmotic/oncotic properties but instead is apparently related to PEG/cell interactions that confer stability during hypothermia” [89].

Finally, due to their valuable insights, we now know that, in these interactions, the role of mitochondria, their integrity, and their function are determinants for hypothermic graft preservation, especially when marginal livers (steatotic and CDC) are used. In summary, Belzer and Southard [1,2] “left the door open for us” to return to the origins of preservation solutions and suggest new paradigms to increase liver cold graft preservation with PEG35 solutions for clinical transplantation purposes.

## Figures and Tables

**Figure 1 ijms-23-05742-f001:**
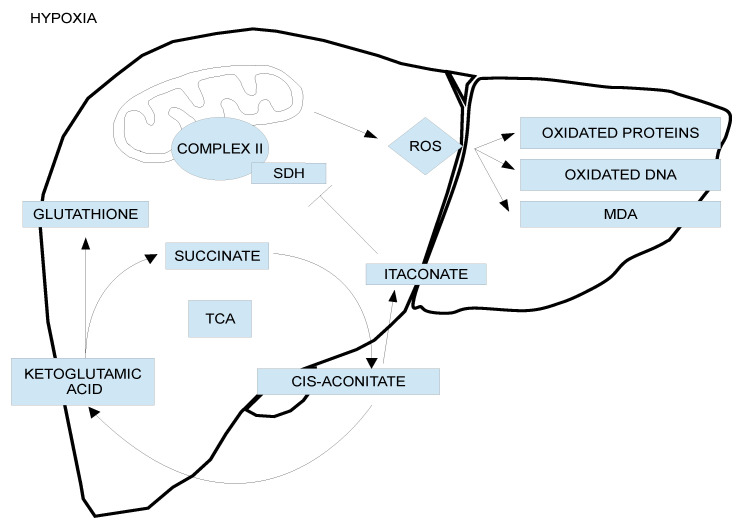
Intracellular mechanisms of ischemic injury. The lack of oxygen to the preserved graft during cold storage provokes a mitochondrial switch to anaerobic metabolism with the interruption of electron flow and mitochondria machinery, the accumulation of interacting energy metabolites succinate and itaconate, and rapid adenosine-triphosphate breakdown. These events that occur during graft cold preservation are modulated by organ preservation solutions. When oxygen is reintroduced under hypothermic oxygenated perfusion and/or with further normothermic conditions, mitochondria re-establish the electron flow with rapid and suitable consumption of the accumulation of succinate at the end of cold ischemia. Subsequent release of reactive oxygen species from complex I occurs. Mitochondria and energy breakdown appear, therefore, as the main targets not only to improve graft quality but also to identify a valid biomarker to predict the “graft status” after static cold storage in organ preservation solutions just before transplantation procedures.

**Figure 2 ijms-23-05742-f002:**
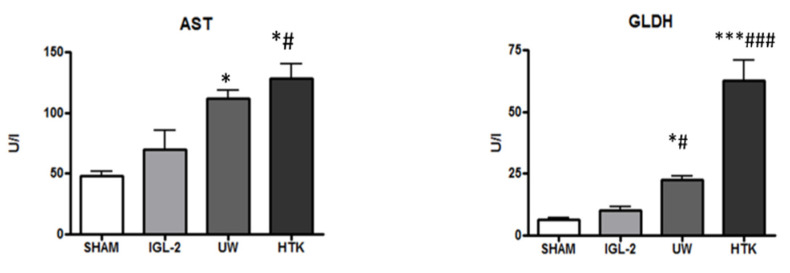
Liver injury and mitochondrial integrityof fatty liver graft preserved in IGL-2, UW, and HTK solutions (24 h; 4 °C). The bars represent the mean values ± SEM of each group (n = 4–6). Differences are shown comparing groups (* vs. Sham, # vs. IGL-2) according to the one-way ANOVA test and the Tukey post-hoc test (one symbol indicates *p* < 0.05; three symbols indicate *p* < 0.001).

**Figure 3 ijms-23-05742-f003:**
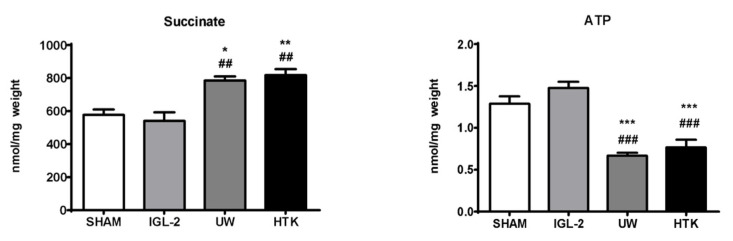
Succinate accumulation and ATP breakdown in fatty livers preserved in IGL-2, UW, and HTK solutions (24 h; 4 °C). The bars represent the mean values ± SEM of each group (n = 4–6). Differences are shown comparing groups (* vs. Sham, # vs. IGL-2) according to the one-way ANOVA test and the Tukey post-hoc test (one symbol indicates *p* < 0.05; two symbols indicate *p* < 0.01; three symbols indicate *p* < 0.001).

**Figure 4 ijms-23-05742-f004:**
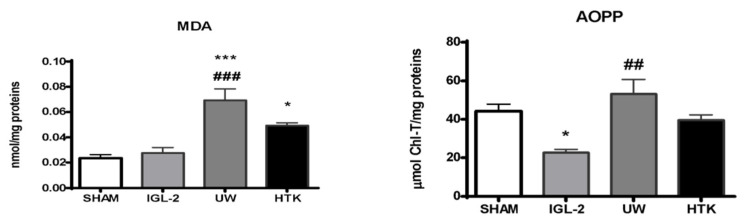
Lipid peroxidation (measured as MDA levels) and oxidized proteins (AOPP) in liver grafts in IGL-2, UW, and HTK solutions (24 h; 4 °C). The bars represent the mean values ± SEM of each group (n = 4–6). Differences are shown comparing groups (* vs. Sham, # vs. IGL-2) according to the one-way ANOVA test and the Tukey post-hoc test (one symbol indicates *p* < 0.05; two symbols indicate *p* < 0.01; three symbols indicate *p* < 0.001).

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
