# Peer review of "Liver Graft Hypothermic Static and Oxygenated Perfusion (HOPE) Strategies: A Mitochondrial Crossroads"

_ijms, 2022, doi:10.3390/ijms23105742_

Round 1
Reviewer 1 Report
The submitted review is a very short one in terms of cited works (only 49 references). It must be elaborated more to be considered for publication in a prestigious journal such as IJMS. Besides, the reason behind writing this mini-review is not stated. Some parts of it are indeed interesting but it leaves the reader nor completely satisfied. Some suggestions are presented below.
Adding a continuous line numbering would greatly facilitate the review process..
Page 1, “All these questions” – exactly what questions?
Page 1, “when he decided to ship a recovered kidney in San Francisco (USA)
to Leiden transport for transplanting” please state the year when it happened.
Page 2, please provide the composition of the preservation solutions.
Page 2, “Nowadays, the accepted critical ischemic time (CIT) for clinical liver transplantation is 12-hours; however, CIT≥4h is associated with considerably lower graft survival than CIT<4h.” please provide some numbers and examples to support this statement.
Figure 1 must be significantly improved. Do the colors have any meanings? What the arrows actually mean?
Page 3, why the exact compositions of the solutions are not provided nor compared?
Page 4, “It is well known that the fatty livers show a reduced tolerance for ischemic events with further reduced ATP levels and greater injury level compared to non-steatotic livers” – it may be well known but it still requires some references.
Page 4, what are the other inhibitors of UCP-2? Are they used in clinics?
Page 5, the caption of Figure 4 completely covers the Figure. Besides, the editorial level of this review is poor.
Page 5, I may agree but what about the oxidant properties of O2 and especially O3?
Part “4.”-what about the other antioxidants?
Page 6, please define “AICAR”, present its molecular structure as well as the one of resveratrol. Again, what about the other AMPK inducers?
Author Response
Good morning, first of all we would like to appreciate the time and patience to review this manuscript. We hope we adressed all the points suggested by the reviewer. Since we can't upload more than one document at once, we adress the suggestions of the reviewer here and we upload the manuscript indicating in red the changes done. And, again, thank you very much for your time.
REVIEWER 1
The submitted review is a very short one in terms of cited works (only 49 references). It must be elaborated more to be considered for publication in a prestigious journal such as IJMS. Besides,the reason behind writing this mini-review is not stated . Some parts of it are indeed interesting but it leaves the reader nor completely satisfied. Some suggestions are presented below. Adding a continuous line numbering would greatly facilitate the review process.
We are grateful for your excellent revision of our manuscript. Thanks for your valuable comments, suggestions and time. We have extended the citations to increase the interest of the readers.
The main reasons for writting the present manuscript were focussed in the recent investigations done by Andrea Schlegel et al. and others [25] who defined that the basis of protective dynamic HOPE strategies are lied to the mitochondrial integrity and machinery function. Please, see Ref 25.
In this manuscript we stated that the proposed mitochondrial mechanisms in HOPE by Schlegel et al. [Ref 25] are the same as those that occur in hypothermic static conditions, which are characterized by the succinate accumulation at the end of cold ischemia. On the contrary, in HOPE, the transient oxygenation of the liver graft is reponsible for maintaining the mitochondrial machinery working at basal levels, reducing thus, the accumulation of succinate. Along this review, we reveal how the key role of complex II and its modulaton are determinant to preserve liver graft in both static and dynamic hypothermic strategies in solid organ transplantation. Following your valuable comments and considerations, the manuscript was corrected and new insighths on mitochondrial markers were proposed and discussed.
-
[Ref 25]: Schlegel A, Muller X, Muller M, Stepanova A, Kron P, de Rougemont O, et al. Hypothermic perfusion protects from mitochondrial injur before liver transplantation. EBioMed 2020.https://doi.org/10.1016/j.ebiom. 2020.103014.
Our answers to your comments and considerations are indicated, as follows:
Page 1, “All these questions” – exactly what questions?.
Sorry, our comments were certainly confuse. “All these questions” are referred to the efforts carried out by Prof. Forket O. Belzer FO to solve the complex problems and circummstances of kidney transportation from San Francisco (USA) to Leiden (Netherlands) in Christmas 1971 for assuring the best graft preservation until that kidney transplantation was succesfully carried out. Please see in Page 2.
We rephrased the sentence according to your suggestions. Ref 1 was added to References section.
-
[Ref 1] [Forket O. Belzer. Organ preservation: a personal perspective. Early experience in kidney transplantation.
https://web.stanford.edu/dept/HPST/transplant/html/belzer.html]
Page 1, “when he decided to ship a recovered kidney in San Francisco (USA) to Leiden transport for transplanting”. please state the year when it happened.
Thanks for your comment. We added the year : 1971, as previously commented. Please see page 2
-
[Ref 1] [Forket O. Belzer. Organ preservation: a personal perspective. Early experience in kidney transplantation.
https://web.stanford.edu/dept/HPST/transplant/html/belzer.html]
Page 2 please provide the composition of the preservation solutions
The composition of the different preservations solutions are shown in Refs 8, 45 and 82. This will facilitate the comprehension to the readers. Thanks.
-
Zaouali, M.A.; Ben Abdennebi, H.; Padrissa-Altés, S.; Mahfoudh-Boussaid, A.; Rosello-Catafau, J. Expert Opin. Pharmacother.,2010, 11(4), 537-555.
-
Bardallo, R.G.; Company-Marin, I.; Folch-Puy, E..; Roselló-Catafau, J.; Panisello- Roselló, A. Carbonell,T. PEG35 and Glutathione Improve Mitochondrial Function and Reduce Oxidative Stress in Cold Fatty Liver Graft Preservation. Antioxidants 2022, 11(1), 158.
[82] Da Silva, R.T.; Bardallo, R.G.; Folch-Puy, E.; Carbonell, T.; Palmeira C.M.; Fondevila, C.; Adam, R.; Rosello- Catafau, J.; Panisello- Roselló, A. IGL-2 as a Unique Solution for Cold Static Preservation and Machine Perfusion in Liver and Mitochondrial Protection. Transplant. Proc. 2022, 54(1), 73-76.
Page 2 “Nowadays, the accepted critical ischemic time (CIT) for clinical liver transplantation is 12-hours; however, CIT≥4h is associated with considerably lower graft survival than CIT<4h.” please provide some numbers and examples to support this statement.
Thanks for your comments. The sentence was newly rephrased. Please see page 2. New references were added in References section (see Refs 22 and 23 ). They are the following ones:
[22] Adam, R.; Cailliez, V.; Majno, P.; Karam, V.; McMaster, P.; Caine, R.Y.; O'Grady, J.; Pichlmayr, R.; Neuhaus, P.; Otte, J.B.; et al. 416 Normalised intrinsic mortality risk in liver transplantation: European Liver Transplant Registry study. Lancet, 2000, 56(9230), 621-627.
[23] Horváth, T.;Jász, DK.;Baraáth, B.; Poles, MZ.; Boros, M.;Hartmann, P. Mitochondrial consequences of organ preservation techniques during liver transplantation. Int. J. Mol. Sci. 2021, 22(6), 2816
Figure 1 must be significantly improved. Do the colors have any meanings? What the arrows actually mean?
Figure 1 was done again. We hope this helps improving its quality image. The legend of figure 1 was changed and expanded. Please see in page 3.
Page 3, why the exact compositions of the solutions are not provided nor compared?
Thanks for you comment. The exact composition of solutions has been exhaustively reported in Ref 8, 45 and Ref 82. This will facilitate the comprehension to the readers. Thanks.
[8] Zaouali, M.A.; Ben Abdennebi, H.; Padrissa-Altés, S.; Mahfoudh-Boussaid, A.; Rosello-Catafau, J. Expert Opin. Pharmacother.,2010, 11(4), 537-555.
[45] Bardallo, R.G.; Company-Marin, I.; Folch-Puy, E..; Roselló-Catafau, J.; Panisello- Roselló, A. Carbonell,T. PEG35 and Glutathione Improve Mitochondrial Function and Reduce Oxidative Stress in Cold Fatty Liver Graft Preservation. Antioxidants 2022, 11(1), 158
[82] Da Silva, R.T.; Bardallo, R.G.; Folch-Puy, E.; Carbonell, T.; Palmeira C.M.; Fondevila, C.; Adam, R.; Rosello- Catafau, J.; Panisello- Roselló, A. IGL-2 as a Unique Solution for Cold Static Preservation and Machine Perfusion in Liver and Mitochondrial Protection. Transplant. Proc. 2022, 54(1), 73-76.
Page 4 “It is well known that the fatty livers show a reduced tolerance for ischemic events with further reduced ATP levels and greater injury level compared to non-steatotic livers” – it may be well known but it still requires some references.
Thanks for the relevance of your considerations. We added some additional references to reinforce the reduced tolerance of steatotic livers vs non steatotic ones against cold ischemia insult. Please, see comments and Refs. 20 and 21 in page 2.
[ 20 ] Panisello-Roselló A, Alva N, Flores M, Lopez A, Castro Benítez C, Folch-Puy E, Rolo A, Palmeira C, Adam R, Carbonell T, Roselló-Catafau J. Aldehyde Dehydrogenase 2 (ALDH2) in Rat Fatty Liver Cold Ischemia Injury. Int. J. Mol Sci. 2018, 19(9), 2479.
[21 ] Zaoualí MA, Reiter RJ, Padrissa-Altés S, Boncompagni E, García JJ, Ben Abnennebi H, Freitas I, García-Gil FA, Rosello-Catafau J. Melatonin protects steatotic and nonsteatotic liver grafts against cold ischemia and reperfusion injury. J Pineal Res. 2011,50(2), 13-21.
Page 5, what are the other inhibitors of UCP-2? Are they used in clinics?
Other UCP inhibitors for reducing UCP2 accumulation and comments have been included in the text. Please, see page 5 and Refs 42 and 43, in References section. To our knowledge, no clinical applications have been carried out. Thanks for your valuable comments.
[42] Petrenko, A.Y.; Cherkashina,,D.V.; Somov,A.Y.; E.; Tkacheva, E.N.; Semenchenko, O.A.; Lebedinsky, A.S.; Fuller, B.J. Reversible mitochondrial uncoupling in the cold phase during liver preservation /reperfusion reduces oxidative injury in the rat model. Cryobiology , 2010, 60, 293–300.
[43] Rial,E.; Rodríguez-Sánchez,L., Aller, P.; Guisado,A.; M Mar González Barroso ,M.M.; Gallardo-Vara,E.; Redondo-Horcajo,M; Castellanos,E.; Fernández de la Pradilla R.; Viso, A. Development of chromanes as novel inhibitors of the uncoupling proteins. Chem Biol. 2011, 18(2):264-74.
Page 5, the caption of Figure 4 completely covers the Figure. Besides, the editorial level of this review is poor.
Sorry for the deficiencies of the text. This was corrected. We evidenced these editorial pitfalls after submission. We believe it is due to low compability of the different text processors used (Microsoft office and open office). Thanks for your personal patience, tolerance and time. We will make sure we correct it.
Page 5, I may agree but what about the oxidant properties of O2 and especially O3?
Ozone has shown to have antioxidant properties [Refs 49, 50] Also, in livers subjected to IRI, the ozone regulates ATP levels as a result of its oxidative effect that may be protective against liver ischemia-reperfusion injury. Moreover, it is thought that it contributes positively to ATP levels and adenosine, xanthine and hypoxanthine metabolism in hepatic ischemia [Refs 50, 52]. See additional comments in Pages 6-7.
[50] Peralta C, Leon OS, Xaus C, et al. Protective effect of ozone treatment on the injury associated with hepatic ischemia reperfusion: antioxidant-prooxidant balance. Free Radic Res. 1999;31(3), 191-196.
[52] Peralta C, Xaus C, Bartrons R, Leon OS, Gelpi E and Rosello-Catafau, J. Effect of Ozone Treatment on Reactive Oxygen Species and Adenosine Production During Hepatic Ischemia-Reperfusion. Free Radic Res. 2000, 33, 5, 595-605,
The deletereous oxidation consequences from oxygen due ozone during cold storage would be presumably negligibles, even more when glutathione is present in the composition of UW solution. An additional comment was included in this page 6.
Page 7 -what about the other antioxidants?
An additional paragraph as ” Other antioxidants “ was added. See Page 6.
We would like to remark the high lability of gluthahione molecule in organ preservation solutions is a relevant limitation as described Boudjema et al. [53] and van Breussegem A et al. [54], respectively, See comments in page 7.
[53] Boudjema K, Van Gulik TM, Lindell SL, Vreugdenhil PS, Southard JH, Belzer FO.Effect of oxidized and reduced glutathione in liver preservation.Transplantation. 1990, 50(6):948-5
[54] van Breussegem A, van Pelt J, Wylin T, Heedfeld V, Zeegers M, Monbaliu D, Pirenne J, Vekemans K. Presumed and actual concentrations of reduced glutathione in preservation solutions.Transplant Proc. 2011, 43(9):3451-4.
In addition, the use of other antioxidants as vitamine C and N-aceytylcisteine (NAC) as additives in preservation solutions was also commented and reported. Please, see Refs 55-58 incorporated to References Section.
[55] Aneta Ostrózka-Cieslik.The Effect of Antioxidant Added to Preservation Solution on the Protection of Kidneys before Transplantation. Int J Mol Sci. 2022,23(6), 3141.
[56] Mehmet Haberal 1, Mahir Kirnap, S Remzi Erdem, B Handan Ozdemir, K Michael Lux, Didem Bacanli.Evaluation of New Baskent University Preservation Solution for Kidney Graft During Cold Ischemia: Preliminary Experimental Animal Study. Exp. Clin. Transplant, 2019, 17(3), 287-297.
[57] Aliakbarian M, Nikeghbalian S, Ghaffaripour S, Bahreini A, Shafiee M, Rashidi M, Rajabnejad Y. Effects of N-Acetylcysteine Addition to University Wisconsin solution on the Rate of Ischemia-Reperfusion Injury in Adult Orthotopic Liver Transplant. Exp Clin Transplant. 2017, 15(4):432-436.
[58] C J Baker , J Longoria, P V Gade, V A Starnes, M L Barr. Addition of a water-soluble alpha-tocopherol analogue to University of Wisconsin solution improves endothelial viability and decreases lung reperfusion injury. J Surg Res.,1999, 86(1):145-149.
Page 6, please define “AICAR”, present its molecular structure as well as the one of resveratrol. Again, what about the other AMPK inducers?
Following your considerations, we define the molecular structures of AICAR and resveratrol as shown in Page 7. However, the use of other AMPK inducers, different to AICAR, added to preservation solution is very poor. On this line, metformin was added to UW solution for hypothermic machine perfusion purposes. Please see page 7 and [Ref 63] in References section.
Other natural AMPK activators, as resveratrol and others could be considered candidates for future approaches. See Refs [64 -65] in References section.

Reviewer 2 Report
The manuscript titled "LIVER GRAFT HYPOTHERMIC STATIC AND HOPE ..." by Raquel Gomez-Bardallo and others is quite a good work, which can be recommended for publication with minor corrections, mainly technical.
Below is my list of proposed changes.
1. Please change the title somehow, the ending sounds strange.
2. I don't know if there are too many keywords?
3. The last two sentences in the abstract - are they necessary - especially the last one?
4. Fig 1 - please improve its quality.
5. Why are there two dots? "2.. Cold ischemic ...",
6. Fig 2. - somehow tragic, please correct it.
7. Why are all these figures so shuffled? Can't the authors see it?
8. Fig 4 - it is a total world championship - fingers enter the text !!! How did the authors do the pdf, they did not see it? I have not seen such carelessness in a long time. Please correct it.
9. Summary was written very briefly - it is entirely correctable.
10. References - terribly carelessly formatted. As for a review article, there should be many more of them here. Especially since the authors gave a lot of keywords.
To sum up - the work is really nice, but very carelessly. Do the authors know what IF has this journal ???
I would like to see the revised works.
Author Response
Good morning, first of all we would like to appreciate the time and patience to review this manuscript. We hope we adressed all the points suggested by the reviewer. Since we can't upload more than one document at once, we adress the suggestions of the reviewer here and we upload the manuscript indicating in red the changes done. And, again, thank you very much for your time.
REVIEWER 2
The manuscript titled "LIVER GRAFT HYPOTHERMIC STATIC AND HOPE ..." by Raquel Gomez-Bardallo and others is quite a good work, which can be recommended for publication with minor corrections, mainly technical.
Thanks for your patience, consideration and time. With our sincere gratitude to correct the manuscript, comments and considerations.
Below is my list of proposed changes:
1. Please change the title somehow, the ending sounds strange.
Thanks for your comments. The title was changed.
2. I don't know if there are too many keywords?
Key words were reduced according to your suggestion.
3. The last two sentences in the abstract - are they necessary - especially the last one?
The abstract was extended and corrected. The last two sentences were omitted and the abstract was changed and re-writen, following your comments and considerations.
4. Fig 1 - please improve its quality.
The quality of Figure 1 was done again and uploaded in a different format. We hope this helps to improve its quality. Foot of the figure was changed and corrected.
5. Why are there two dots? "2.. Cold ischemic ...",
Sorry again once. The pitfall was corrected.
6. Fig 2. - somehow tragic, please correct it.
Absolutely agree. Sorry again once.Thanks for your tolerance.
7. Why are all these figures so shuffled? Can't the authors see it?
We had format problems not detected. We are working with different text processors (microsoft office and open office) and their compatibility sometimes seems not to be working.
-
Fig 4 - it is a total world championship - fingers enter the text !!! How did the authors do the pdf, they did not see it? I have not seen such carelessness in a long time. Please correct it.
Sorry again. We believe it is due the low compatibility among text processors. Somehow we did not see it in the pdf version. We will make sure that this editting issue won't happen again.
9. Summary was written very briefly - it is entirely correctable.
We agree with you. The abstract was corrected and extended.
10. References - terribly carelessly formatted. As for a review article, there should be many more of them here. Especially since the authors gave a lot of key words.
References section was correctly formatted and extended(90 citations).
To sum up - the work is really nice, but very carelessly. Do the authors know what IF has this journal ???.I would like to see the revised works.
Sorry for the chaotic presentation of the manuscript. We corrected them in the revised version. In the uploaded version we did not notice the deviance in the format, but we think it is because of the low compability among the different text processors. We will make sure hat this does not happen again. Now we are aware of the issue.

Round 2
Reviewer 1 Report
I can see that the Authors did their best to improve the manuscript. While I am not entirely convinced about it, I appreciate their effort and I think the manuscript can be possibly accepted for publication.
Reviewer 2 Report
The authors have improved the manuscript very well - I think it is acceptable now.